# The Use of Infrared Thermography to Develop and Assess a Wearable Sock and Monitor Foot Temperature in Diabetic Subjects

**DOI:** 10.3390/s21051821

**Published:** 2021-03-05

**Authors:** José Torreblanca González, Beatriz Gómez-Martín, Ascensión Hernández Encinas, Jesús Martín-Vaquero, Araceli Queiruga-Dios, Alfonso Martínez-Nova

**Affiliations:** 1School of Industrial Engineering, University of Salamanca, E37700 Salamanca, Spain; torre@usal.es (J.T.G.); queirugadios@usal.es (A.Q.-D.); 2Department of Nursing, Centro Universitario de Plasencia, University of Extremadura, E10600 Plasencia, Spain; bgm@unex.es (B.G.-M.); podoalf@unex.es (A.M.-N.); 3Faculty of Sciences, University of Salamanca, E37008 Salamanca, Spain; ascen@usal.es

**Keywords:** diabetic foot, gait, monitoring foot temperature, smart wearable

## Abstract

One important health problem that could affect diabetics is diabetic foot syndrome, as risk of ulceration, neuropathy, ischemia and infection. Unnoticed minor injuries, subsequent infection and ulceration may end in a foot amputation. Preliminary studies have shown a relationship between increased skin temperature and asymmetries between the same regions of both feet. In the preulceration phase, to develop a smart device able to control the temperature of these types of patients to avoid this risk might be very useful. A statistical analysis has been carried out with a sample of foot temperature data obtained from 93 individuals, of whom 44 are diabetics and 49 nondiabetics and among them 43% are men and 57% are women. Data obtained with a thermographic camera has been successful in providing a set of regions of interest, where the temperature could influence the individual, and the behavior of several variables that could affect these subjects provides a mathematical model. Finally, an in-depth analysis of existing sensors situated in those positions, namely, heel, medial midfoot, first metatarsal head, fifth metatarsal head, and first toe has allowed for the development of a smart sock to store temperatures obtained every few minutes in a mobile device.

## 1. Introduction

### 1.1. Diabetic Foot

Diabetic foot syndrome is defined as the infection, ulceration or destruction of the deep tissues of the foot, associated with neuropathy and/or peripheral vascular disease of different magnitude, in the lower extremities of patients with diabetes mellitus [1]. The incidence of foot ulcers in diabetics rounds between 15 and 25% [2] and is a frequent cause of hospitalization and could lead to major complications, like lower limb amputations [3]. Actually, it is estimated that about 85% of diabetics suffering from amputations have previously had an ulcer [4]. The mortality rate in subjects with diabetic foot syndrome is more than twice as high than an average population [5].

### 1.2. The Role Temperature, Pressure Points and Activity on Diabetic Foot

The human being is homeothermic, that is, it maintains the central body temperature constant (oscillating between $36.5{\;}^{\circ }$C and 37.2 ${}^{\circ }$C) despite the variations of ambient temperature. Human beings control their temperature by thermoregulation, where the skin, as the body’s largest organ, is a key factor in this process [6]. Skin temperature, in the normal human being, is controlled through many different mechanisms; especially in the extremities (fingers and toes) microcirculatory vasomotion is a crucial determinant of heath preservation of release (not sweating as stated, although being just another way—ischemia and necrosis does not result from sweat gland impairment), and, also, it is normally dependent of the room conditions.

During an activity such as walking, an increase in internal heat is generated, which is manifested in a similar way in the increase in temperature of the skin of the feet [7]. In addition, these are integrated into a complex sports sock-shoe that makes it difficult to transpire properly and evacuate the temperature generated. Thus, socks, together with footwear, become an important element, not only to protect the skin from injury but also to control thermal conditions [8]. In the same way, there are fundamental pieces in the control of moisture (as it will act in the transport of heat in the skin) and therefore in ensuring a correct hydration of the foot [9].

Hence, the evaluation of the pruning temperature helps one to know the internal conditions, which could help to prevent lesions associated with the temperature during gait, or that manifest with changes of the same. An increase of the foot temperature would generate an excess of transpiration in addition to generate changes in the pH of the skin that can turn the foot into the breeding ground for bacterial infections. Similarly, a sudden increase in temperature of an area relative to its contralateral may be indicative of a high risk of injury development, although a reduction may indicate a risk of ischemia. These alterations could lead to further complications, such as pain during gait and development of ulcers, that can become a serious problem in diabetic subjects, with a serious risk of amputation.

Thermographic evaluation of the sole surface of the foot is particularly important in the studies of pathologies associated with the foot at risk, either from neuropathy or from peripheral vasculopathy. Thus, it has been possible to determine certain asymmetries, such as an increase in temperature of $2.2{\;}^{\circ }$C in an area relative to its contralateral, which indicates an underlying subclinical inflammation without apparent signs [10]. This could be a determinant of the risk of ulceration in this area. The sole thermographic evaluation is carried out in different regions of interest (ROIs), being very variable in number and location. The choice of areas of interest can be of great importance, as it could relate the increase or decrease in temperature to the risk of injury in that area, such as a plantar ulcer. However, the literature offers numerous studies, with disparity in number, location and reasoning for choosing ROIs. Recent literature has found studies that analyze from 4 [11,12] to 12 [10] zones, with a number of 5–6 being the most common.

The researchers seem to agree on the study of four specific areas: heel, inner forefoot, fifth metatarsal head and first finger. However, there are important differences, Astasio-Picado et al. [11] analyzed the first metatarsal head, while Chatchawan et al. [13] and Bagavathiappan et al. [14] extended the area to include also the second head. Other researchers focused their attention on five forefoot areas, namely 1st, 3rd and 5th metatarsal heads and 1st and 4th toes [15]. Similarly, Gatt et al. conducted a study in 8 forefoot, medial, lateral, central and the toes [16].

However, other studies do not specify the exact number of regions or their location [17,18], where there is no clear consensus on the criteria for choosing the areas studied, since the studies do not specify this criterion. Thus, it appears that the choice of these areas may be related to areas of frequent ulcer occurrence [11], but in others, the criteria are not specified either. Thus, in neuropathic feet, the highest prevalence was found in the fingers (40.4%) and in the metatarsal area (39.1%), while the ischemic foot group is the most frequent area in the fingers, up to 63.6%. On the other hand, the neuro-ischemic foot group (frequent alteration in diabetic feet of time of evolution) the distribution of ulcers was 51.8% in the area in the inner metatarsal area and the fingers, mainly the first [19].

It seems that the inclusion of the areas of first metatarsal head and first finger is highly recommended, for around 50% of combined prevalence of ulcer appearance. However, the other 50% occur in different areas of the foot, with a significantly lower prevalence, so monitoring other areas becomes a necessity. Choosing areas that are representative of the risk of injury but also able to discriminate between homogeneous temperature zones and provide data on the whole foot without subdividing it into too many regions would be of great importance for rapid realization, simple and reliable podiatric screenings that evaluate the risk of a diabetic foot.

### 1.3. The Utility of Wearables in Detecting Temperature and Aims of the Study

The proposal of this research is to develop a system, a smart sock, capable of measuring temperature at various points on the foot, to record these measurements during gait by using a smartphone and finally to analyze the data and alert the patient where necessary, i.e., a remote health monitoring system [20,21].

The first step was to make a prediction model so that when any of the measurements exceeds a certain value, the smart sock will send an alert to the telephone and the patient will know that he/she should stop because there is a problem. To develop the mathematical model, data have been collected from a group of diabetic and nondiabetic individuals and a statistical analysis has been developed.

The objective of this work is to analyze and provide good reasons about the number and location of ROIs, which are necessary to perform a good screening of the diabetic foot and to be able to optimize the study, adding necessary areas or eliminating others that offer redundant results. Moreover, a detailed description of the sensors that will be used to measure foot temperate is also included in this study. Thus, the layout of the paper is as follows: A brief overview of the most common sensors employed to measure the temperature is given in Section 2 and their main features in relation to the main goal of the paper are analyzed. In Section 3, a survey that conducted to study the most important variables to study the temperature in both feet is described. A basic statistical analysis of the data is provided and several graphs of feet temperature to determine the most important ROIs are showed. Through further analysis of the sample, a relation of the feet temperature with some of the other variables was found, which allows to develop the corresponding model. The main part of this work is described in Section 4, where a new prototype for a smart sock is described. This smart sock is able to continuously obtain temperatures at several points of the foot, this leaves open the possibility of advancing the study of this disease in the near future. Finally, some conclusions and goals are given in Section 5.

## 2. Types of Sensors to Monitor Temperatures

Temperature can be defined as a physical quantity that shows the amount of heat in a body. Its perception is linked to the notion of cold and heat.

Its measurement is carried out using a temperature sensor, an instrument that collects the temperature data from a certain source and converts it into information that is understandable by a device or an observer. As the Electronics Tutorials website points out, temperature sensors can be classified into two main groups [22]:Contact sensors, which must physically touch the object, using conductivity to measure changes.Noncontact sensors, which use convection and radiation to warn of a change in temperature.

Of these two types of sensors, the most interesting in this research are the first ones in which there is contact with the object or patient to be measured. The temperature of the human body has been measured and taken into account from the very beginning of medicine. Things related to the change in temperature in different areas of the body have been discovered and have come to the study of numerous diseases.

Measurement taking of this variable is, currently, very well resolved for measurements in industrial processes and for many areas in the human body, but perhaps it is not as well resolved for measurement on the sole of the foot. Some authors have developed an insole [12,23], but our goal is to define the characteristics of a sock that will help to reliably control foot temperature.

Focusing on the contact sensors and, in addition, that they must be small to prevent injuries, the following possibilities were found [24]: Thermocouples, thermoresistances, thermistors, diodes and programmable electronic devices. Detailed information about these electrical devices can be found in [25].

Other sensors are not adequate because it is complicated to obtain a magnitude (temperature) every few minutes when a reliable value is needed. So, infrared, mechanical, color change sensors, etc., are the most suitable for the purposes of this study.

One type of devices that has been tested were thermocouples. Due to its characteristics related to cost, size and temperature range, between others, thermocouples are considered the workhorse of devices capable of measure temperature. A detailed description of different type of thermocouples could be found in [26,27].

Thermoresistances work by varying its resistance with temperature. Their sensitive elements, based on metallic conductors, change their electrical resistance depending on the temperature. The most common devices (called PT100, PT1000, etc.) are built with a platinum resistance. These resistance temperature detector works over a range that varies between $-200^{\circ }$ and $800^{\circ }$ with only a single calibration point, and it is considered the best accuracy tool [28].

Thermistors are much more sensitive [29], made up of a synthesized mixture of metal oxides. They are essentially semiconductors that behave like “thermal resistors”. They can be found on the market with the denomination NTC (negative temperature coefficient, i.e., the resistance decreases with temperature) and PTC (positive temperature coefficient, i.e., it increases resistance with temperature). They are much easier to measure than thermocouples and thermoresistances, as a simple voltage divider is enough to get the temperature [30,31].

Finally, diodes base their operation on the voltage variation in their terminals since it depends on both the current flowing through it and the temperature at which the diode is located; this variation occurs at the PN junction, very sensitive to temperature changes depending on the internal doping they have. This internal doping makes the internal resistance of the PN junction vary as a function of temperature, which causes the current to increase or decrease and by Ohm’s law the same occurs with the voltage at the semiconductor ends. There are a wide variety of devices to measure temperature, from those that vary its voltage (diodes), those that give gradual values of voltage as a function of temperature (analog integrated circuits) to those that are programmed and give a sequence of bits to obtain the value of the temperature (digital integrated circuits). LM35 is an example of analog integrated circuit [32].

The latest generation of sensors for temperature measurement are integrated circuits that, in addition to measuring temperature, can communicate with microcontrollers. The temperature variation is done electronically, as diodes, by variation of voltage and current in the PN junction of the semiconductors. Some examples of these sensors are the MAX30205, the Si7006, etc. [33,34,35,36].

From all these sensors, the ideal ones are the programmable electronic devices in integrated circuits, due to their great versatility in being able to program. In addition, their electrical connection is easier by having compatible communication lines between all the sensors that are connected at the same time, such as the I2C communication. Another advantage is the two-way communication possibility in such a way that the temperature value can be obtained directly, and it is not necessary to carry out operations with the values read from the device. The electrical connection of these devices can be very simple with three or four wires at most. Its biggest drawback is electrical welding as they must be made with special materials and equipment, due to the small size of the electrical connections, which are often in the order of 200 microns.

Another very interesting choice are thermistors, since they are small devices, with less rigidity than the previous ones and they are probably better to be included in a sock. Its biggest drawback as with the previous ones are the electrical connections. At this point, they have a certain advantage over the rest of devices, since being less rigid, they can be welded and arranged in a better way. Another disadvantage compared to the previous ones is that the obtained temperature signal must be treated to finally obtain the temperature value, which makes the entire signal acquisition system more complex.

As for the other sensors, the same problems as with programmable electronic devices and thermistors must be addressed. The first ones, being programmable, they already do everything that is needed, i.e., to acquire the signal and transform it to an understandable temperature value for the user, while the seconds may be easier to handle for electrical mounting but worse for acquiring the signal and transforming it to a value understandable by the user.

In any case, and depending on the type of sensor, it could be complicated to place many sensors (not more than 4 or 5), especially if we also want to measure other variables such as pressure or humidity, which are not in the scope of this paper. Hence, it is necessary to know the most important points where temperatures should be calculated.

## 3. Data Analysis

The goal of this study is to select the positions for sensors to measure the foot temperature and thereby preventing the occurrence of ulcers in diabetic patients. With the aim of analyzing the best position for those sensors, we obtained foot temperatures measured by infrared thermography. A survey with diabetic and nondiabetic individuals was conducted to collect and compare several variables, not only temperatures.

### 3.1. Data Collection

The sample consisted of 93 subjects. All were patients of the CPUEX Clinic of the University of Extremadura. Since CPUEX Clinic is not large, it is considered that they are a good representation of this institution since the number of male and female patients is similar, also the people with and without diabetes. This analysis is expected to continue with more patients from other institutions. In this way, it will be easier to follow the standards suggested in works such as [37], for example, in the number of subjects and the way to choose them. The thermal images of patients that participate in this research were taken with a FLIR E60bx Infrared camera.

The measure of foot temperature was taken before and after a 100 m walk. In this study patients of different ages were considered, some of them elders, and they cannot walk long distances. Moreover, they should not go outside because the measures could be affected by ambient temperature. A 100 m walk was sufficient to cause a change in temperature by activation. In the future, to study how walking longer distances may affect diabetic patients could be considered.

Although some studies attempt to define the most critical ROIs and thus more likely to suffer serious injuries, there is no final decision about that. This is the reason why a total of 17 temperature measurements from 17 ROIs were collected, from the plant and dorsal areas (9 form the plant and 8 form the dorsal), from both feet. These 17 ROIs were selected following anatomical and functional criteria (i.e., importance in gait, blood flow and risk of ulceration). Plantar measurements included the heel, being not included it in dorsal vision. Measures (indices) that have been selected were: from heel $\left(I_{1}\right)$, medial midfoot $\left(I_{2}\right)$, lateral midfoot $\left(I_{3}\right)$, first metatarsal head $\left(I_{4}\right)$, central metatarsal heads $\left(I_{5}\right)$, fifth metatarsal head $\left(I_{6}\right)$, first toe $\left(I_{7}\right)$, central toes $\left(I_{8}\right)$ and fifth toe $\left(I_{9}\right)$, and the same areas in the dorsal $(I_{10}$–$I_{17})$ [38].

Data collection was made using a survey that includes 11 sections:General data about the individual, such as gender, age, weight, height, blood type, and the use of high heels; data specific for diabetes patients: type of diabetes, treatment, control of their diabetes; individuals suffering neuropathy disability; results of the modified Edinburgh questionnaire to diagnose arterial claudication; ankle-brachial index (ABI) and measurement of glycosylated hemoglobin.Habits: Way of life (sedentary lifestyle or practicing a sport); type of feeding (healthy diet or not); alcohol consumption; smoker or nonsmoker; any regular medication; and dominant foot.Comfort: Recent activity with current shoes; intensity of the most recent activity (no activity, walking, running, etc. during the last 30 min); comfort level with the current footwear (during the last 30 min) by means of a 1 to 5 Likert scale, being 1 “very comfortable” and 5 “very uncomfortable”; and the type of footwear that they usually use.Blood pressure and central temperature: data of systolic blood pressure (high), diastolic blood pressure (low), and central temperature (thermometer).Climate data (season data) including date, and number of photos from the foot (sole and dorsal from both feet).Temperature measurements (prewalk right foot data) were performed with a thermal camera FLIR E60bx, with an infrared resolution of 320×240 pixels, a sensitivity of $0.05{\;}^{\circ }$C and a precision of ±2%. Thermal picture were taken 1 m far from a foot covered in black cardboard. A total of 17 data average temperatures (9 plantar and 8 dorsal) of each participant were collected and stored using the The Flir Tools${}^{®}$ software (Similar to Figure 1). This number of ROIs has been used previously in thermographyc assessments [38,39,40] (see Figure 2). Room humidity (%) and temperature (${}^{\circ }$C) were also measured with a Flir MR 77 device.Temperature measurements (prewalk left foot data).Temperature measurements (postwalk right foot data). After the 100 m walking, the temperature was recorded in the same way as in point 6.Temperature measurements (postwalk left foot data).Foot Posture. Scores of Foot Posture Index (FPI) for left and right foot were assessed following standard procedure [41], subjects in their relaxed stance position, both limb support, arms relaxed and looking straight ahead; foot posture classification was found to be neutral when the score was between 0 to 5, supinated from −1 to −12 and pronated from 6 to 12 [42]; presence of plantar hyperkeratosis at any zone of the foot was also recorded.Final comments and remarks.

Each measurement take the research team a set-up of 20 min for each participant.

A study with a sample of 93 individuals was carried out, of whom 44 are diabetics and 49 nondiabetics and among them 43% are men and 57% are women. Figure 3 left shows that in both cases (diabetics and nondiabetics, and men and women) the percentage is quite similar, the data is similarly distributed. Histograms of age, weight, height and body mass index (BMI) have also been represented in the right part of Figure 3. The weight basically is between 60 and 90 kg and the height between 1.60 and 1.70 m. The largest number of individuals corresponds to an age between 70 and 80 years since the patients who go to the consultation are older people who have problems for their basic care and the body mass index is between 20 and 35.

The study was conducted in accordance with the Declaration of Helsinki, and the experiment developed for this paper received a positive report from the bioethics and biosafety commission of the University of Extremadura (with Ref. 04/2018). It follows Spanish and European legislation, and all the people who participated in the survey gave their consent for research purposes.

### 3.2. Statistical Data Analysis

Correlation shows the relationship between two variables, whilst regression analysis generates a mathematical equation that serves to predict the behavior of the process output by changing its inputs. Correlation is usually the first analysis carried out since you want to check if there is a relationship between the variables, and, the regression analysis usually takes data in order to find a relationship that provides for the output of the process by changing the data that affect this. In this work these two statistical procedures were used and also dendograms to determine which are the locations where sensors should be placed.

A correlation higher than 0.8 between sole and dorsal indices at the same position was established in [38], so sole data have only been considered (from $I_{1}$ to $I_{9}$ from sole) before and after a 100 m walk. This walk was performed to determine if the diabetic feet, after the walk, show any difference with the nondiabetic feet. To do this different measures were taken, which will allow developing a model with the aim of predicting the temperature in some points of the foot and check if they are within “normal” values or if there is any deviation.

Specifically, the data obtained from the survey detailed in Section 3.1 have been considered.

Dendograms were represented for diabetic and nondiabetic individuals to determine if there is a difference between them and also what are the most influential indices (to select these points as the places to put sensors on the socks). In all dendrograms, the most significant points are $I_{1}$$I_{2}$, $I_{3}$ or $I_{7}$, and points $I_{4}$, $I_{5}$ and $I_{6}$ related to each other, as can be seen in Figure 4 for the right floor of diabetic patients (left) and nondiabetics (right) before the walk.

According to the literature, the points where the most ulcers appear are in the metatarsal area of the foot, i.e., indices $I_{4},I_{5}$, and $I_{6}$ from Figure 2a, and index $I_{7}$ (30% of the ulcers according to [39] in the fingers. The dendrogram indicates that there is a relation between those three metatarsal points. Consequently, indices $I_{4}$ and $I_{6}$ have been chosen because they have a higher percentage, 22% and 11%. Thus, considering points $I_{4}$ and $I_{6}$, all the variability of the upper metatarsal part were obtained.

On the other hand, the index $I_{1}$ was also selected because there is a relation between and there are usually more difficult problems. Apart from that it seems reasonable to consider $I_{1}$, as a greater force is applied to this area when walking and it is a point of greater probability of ulcer. Point $I_{7}$ was also chosen as it is an area not related to the previous ones and presents a probability of ulceration of 30%. The dendogram also indicates that $I_{2}$ is totally different from the rest. The area of $I_{2}$ carries irrigation to the metatarsal head and the first toe. This is an interesting area since if it cools, the front side will also cools and no blood will go there, with the risk of ulcer. Moreover, this point receives less friction from the footwear, with a difference in relation to the other points where we measure the temperature. That is why we decided to include it as it is a singular point, totally different from the others but where practically no ulcers appear. In this way there is an element of control of the other areas that will be considered for the smart sock.

In summary, the points that will be studied are $I_{j}$ with j∈{1,2,4,6,7}.

For this study a pool of candidate variables have been considered, in addition to the temperature. This allowed the possibility of discarding the variables that do not affect diabetic foot. After a stepwise regression, a prediction model has been developed for all the data, in which the following variables are included: AGE, SEX, BMI, DB (Nondiabetic = 1, diabetic = 2), TNMX (Systolic Blood Pressure), TNMN (Diastolic Blood Pressure), TC (central temperature), TEXT (outside temperature), TPRE (air temperature of the room before the walk), HPRE (humidity of the room (in%) before the walk). The coefficients that have been obtained for the different models are shown in Table 1. The model only considering the variables that influence is calculated with the command *Stepwise* provided by the R program.

There are variables that do not influence the model, such as SEX and TEXT, and there are other variables that have little influence, such as BMI. In addition, the HPRE practically only influences before the walk. As it can be observed, being diabetic or not does influence the model, therefore data have been separated in diabetic and nondiabetic people and since sex does not influence the sample was not separated by sex. In Figure 5 the graph of the data for the coefficient SRPRE1 (black, solid line) of the sole before the walk for the complete sample is represented, and a comparison with the complete model (with all the variables, in red, dashed line), and the model with the variables given in the legend (blue, dash-dot line). This model suggests that some of the variables in this study have small influence in the temperatures.

Figure 5 shows the sample data in black color; the blue graph corresponds to the calculated model only considering the variables that influence; finally, the red line is the model when all variables are taken into account. The correlation coefficients obtained correspond to the correlation between the calculated model and the real values (blue r=0.628), between the complete model, considering all the variables and the real values (red r=0.638) and between the blue model and the red model (r=0.985). A confidence interval of 95% was considered for this study. It can be appreciated that both models are very well correlated, so it is not worth using all the variables, only the ones in Table 1. Moreover, the correlation coefficients are not much improved by choosing more variables and it is more realistic to choose only those that influence.

A Bland Altman plot of the model is presented in Figure 6 comparing the two measurements in the index 1 before the walk: the sample data and the model only considering the variables that influence, calculated with the command *Stepwise* provided by R. In other indices, we see similar results. The proposed model is not predictive and the approximation is not quite good, because we have found outliers for both the lower and upper values. The multiple regression coefficient between the real values (SRPRE1) and the model is $R^{2}=0.3948$. As was mentioned in Section 3.1, the study was carried out with all diabetic patients from the podiatry clinic with the goal of selecting the variables that could influence the values of the defined indices and to improve the measurement of these variables.

The goal of this study is the proposal of socks capable to send a signal to a mobile telephone when a difference of temperature is detected, of more than 2 degrees (hyperthermia) or less than 2 degrees (ischemia) between the same indices of the two feet. For doing this the difference of the indices in the sole in the 5 points ($I_{j}$, j∈{1,2,4,6,7}) was calculated and a basic statistical study was developed for all of them starting with all sample data and later on for individuals when diabetics from nondiabetics are separated, and before and after the walk.

As an example, Table 2 shows the data for nondiabetic patients before the walk for the sole of the right foot. In this case the index with the highest coefficient of variation is $I_{6}$, followed by $I_{7}$, in which the interquartile range is also the highest of all. Indices $I_{1}$, $I_{4}$ and $I_{6}$ have a negative skewness and are the ones with the highest kurtosis, while $I_{2}$ and $I_{7}$ have a skewness close to zero and a kurtosis that is also very small.

For the indices $I_{j}$, j∈{1,2,4,6,7}, a violin graph with a boxplot with a mustache inside is represented in Figure 7. It can be appreciated that there is an overlap between the mean (red dot) and the median (middle straight line of the boxplot with a mustache).

## 4. A Prototype of a Smart Sock

### 4.1. Introduction

As was mentioned before, the goal of this research is to get a smart sock, capable of measuring temperature in diabetic foot. Some examples of prototypes of socks appear in [43,44] but sensors there are large, so they are not the better option for diabetic patients. The proposed system will be composed of several sensors located in predefined ROIs, these sensors will be managed by an Arduino board, which will send the collected data to a smartphone that will be able to create an alarm in case of needed. A block diagram of the proposed structure is shown in Figure 8.

### 4.2. NTC TTF 103 Thermistor

In Section 3 the areas of the feet to place the sensors for measure the temperature were selected. The next step is to think about which sensor that could be used to measure the temperature because it is a very delicate situation, since they should not disturb when walking. Thermistors were considered the best option, since they are very sensitive. As was mentioned in Section 2, they can be found on the market with the denomination NTC and PTC.

For the device proposed in this study, the NTC TTF 103 thermistor with 10 kΩ will be used. This sensor has very small dimensions, 25 mm long, 3.8 mm wide and 0.4 mm high, which makes it ideal to avoid disturbing the foot. Moreover, the manufacturer provides tables with the resistive values and their corresponding temperatures. Although data table from the manufacturer is available, they have been calibrated using the Steinhart–Hart equation and their coefficients *A*, *B* and *C* were obtaining.

The Steinhart–Hart equation is an empirical expression that has been determined to be the best mathematical expression for the resistance temperature ratio of NTC thermistors and NTC probe sets. The most common equation is:(1)$$T_{i}=\frac{1}{A+B\ln\left(R_{i}\right)+C\ln^{3}\left(R_{i}\right)},$$
where $T_{i}$ is measured in degrees Kelvin, and *A*, *B* and *C* are calculated following these steps: first of all the thermistor at three different temperatures is measure, and then this values are used to solve the resulting simultaneous equations, considering
$$L_{i}=\ln\left(R_{i}\right),\;Y_{i}=\frac{1}{T_{i}},\;\gamma_{2}=\frac{Y_{2}-Y_{1}}{L_{2}-L_{1}},\;\gamma_{3}=\frac{Y_{3}-Y_{1}}{L_{3}-L_{1}}.$$

Finally, the parameters are obtained taking the following expressions for *A*, *B* and *C*:(2)$$C=\frac{\gamma_{3}-\gamma_{2}}{L_{3}-L_{2}}{\left(L_{1}+L_{2}+L_{3}\right)}^{-1},\;\;\;B=\gamma_{2}-C\left(L_{1}^{2}+L_{1}L_{2}+L_{2}^{2}\right),\;\;\;A=Y_{1}-L_{1}\left(B+CL_{1}^{2}\right).$$

These coefficients are used from three measurements in real conditions. With these three parameters and obtaining the resistance, value the temperature is obtained with the NTC thermistors. It is also possible to obtain that value using the website of Thermistor Calculator for Laser Diode and TEC Controllers. [45].

The six NTC sensors were calibrated (an extra sensor calibration is included, just in case one fail later). As Figure 9 shows, both the resistences that will be placed in series with the NTC thermistor and the thermistor itself are numbered, so considering each resistance together with its NTC, they will always form the same voltage divider.

To carry out the measurements, units calibrated have been used to take the temperature and to measure the resistance. These units are shown in Figure 9. The RS1314 unit has been used for temperature, capable of measuring very precise temperatures with thermocouples. This unit is equipped with two thermocouples that measure the temperature at the same time, and thus they take the values when both measures are the same. To measure the electrical resistance a FLUKE 87 was taken. This device is able to measure various parameters such as resistance, voltage, electric current and it is even possible to measure the temperature with a type *K* thermocouple.

The measurements have been made in four temperature ranges, the first with water almost at $0{\;}^{\circ }$C, specifically at $1.6{\;}^{\circ }$C, the next measurement around $29{\;}^{\circ }$C, and finally the temperature has risen to around $43{\;}^{\circ }$C (we also calculated data at $21.6{\;}^{\circ }$C to check the function given by Equation (Equation 1)). We found that $1.6{\;}^{\circ }$C and $29{\;}^{\circ }$C temperatures were more stable, and $43{\;}^{\circ }$C temperature had some small variations. When we measured both resistance and temperature, these variations have been taken into account when entering data on the website to calculate the parameters of the Steinhart–Hart equation.

The way to obtain these temperatures has been through ice water and waiting about 20 min to stabilize the temperature, then it has been heated with a microwave and cold water has been added to obtain $43{\;}^{\circ }$C and $29{\;}^{\circ }$C. This is shown in Figure 10.

All six NTC sensors have been calibrated. The obtained values of resistance (Res *i*) and temperature (Temp *i*) are shown in Table 3.

Once measurements were taken, the parameters for each thermistor are obtained (see Table 4).

### 4.3. Arduino

The next step is to establish the element for obtaining the measures to dump them on the smartphone. For reading the sensors we started with ARDUINO board (https://www.arduino.cc/, accessed on 13 January 2021).

As is well known, Arduino is a platform of electronics prototypes based on flexible and easy-to-use hardware and open-source software. It is intended for artists, designers, as a hobby and for anyone interested in creating interactive objects or environments.

For this study, an Arduino Nano has been used to collect the data. It is a reduced version of Arduino UNO, although with some differences. Arduino Nano minimizes the energy demand that it consumes and moreover, less space is needed to host the board, making it ideal for this project as it has to be worn as an ankle strap. This Arduino board is a small, flexible and easy-to-use microcontroller. It is based on the ATmega328 microcontroller. It works at a frequency of 16 Mhz. The memory consists of 32 KB of flash memory. It has a 5 V supply voltage, but the input voltage can vary from 7 to 12 V. It has 14 digital pins, 8 analog pins, 2 reset pins and 6 power pins (Vcc and GND). In the case of analogs, they allow a 10-bit resolution from 0 to 5 V. It uses a standard miniUSB for connecting with the computer for programming or power it. Its power consumption is 19 mA. Printed circuit board size is 18 × 45 mm weighing only 7 g.

Arduino microcontrollers have multichannel analog-digital converters. The converter has a resolution of 10 bits, i.e., it takes values between 0 and 1023. Thus, if the resolution is maximum, that is, 5 V, the converter will give an integer value of 1023, if the measured voltage is intermediate, for example 2.5 V, the converter will store an integer value equal to 512, and if the voltage is zero (0 V), then the converter will give an integer value of 0. The Arduino resolution is calculated as the quotient between the reference voltage and $2^{N}-1$. The reference voltage is the maximum voltage that is applied to the converter, normally it is the supply voltage, then 5 V, although it can be modified and changed to a smaller value, thereby increasing resolution. It is not possible to set higher values than the power supply, and lower values can be increased to a certain value threshold set by the manufacturer of the microcontrollers. The exponent *N* refers to the bits resolution of the converter, in this case, 10.

Therefore, the resolution that we have calculated for our device is 4.88 mV.

### 4.4. Smartphone Application

A mobile phone application has been developed to collect and analyze data. This application takes the data from the sensors and transmit them by a Bluetooth connection that is paired with the smartphone. The data transmission is made through the HC-06 Bluetooth module, which only needs to be paired with the mobile in order to receive data. This module only has four terminals, two for power, one to transmit the signal and the last one to receive it, which means that communication can be bidirectional, although in our case we only use unidirectional communication, from the measuring system to the smartphone.

The smartphone stores the data in real time on a plain text file (.csv file) that can then be processed. The phone application has been implemented with App Inventor 2 software. This is developed from a web page available to everyone (https://appinventor.mit.edu/, accessed on 13 Januray 2021).

When building the Android applications we work with two tools: App Inventor Designer and App Inventor Blocks Editor. In Designer environment we build the user interface, choosing and placing the elements with which the user will interact and the components that the application will use. In the Blocks Editor we define the behavior of the components of the application.

In Figure 11 some images of the smart sock working connected to the smartphone application are shown. A set of sensors have been placed in the ROIs established in Section 3, after measuring the temperature data from these regions, they are submitted to the mobile device and stored in a spreadsheet.

On the other hand, a battery called LIPO has been used. It is composed of lithium and polymer, which is a battery widely used due to the large amount of current that it can give at a certain moment, but it is adequate for this proposal because it can be used for a long time without recharging. Nickel-Cadmium batteries and even Nickel-Metal Hydride could be used.

## 5. Conclusions and Future Work

Foot pathologies result directly from several diseases, mainly related with gait, and they are among the most serious and costly complications that affect diabetes mellitus patients. To monitor these patients, their gait and quality care will avoid the increase of costs, patients’ consultation or centers overcrowding. The proposal of this study is to develop a smart sock to monitor diabetic patients’ foot temperature.

We had to solve the following obstacles before this could be achieved:

(i) The first one was that several different papers do not agree about how many sensors are necessary and where they should be placed. To select the ROIs we have analyzed temperature data obtained with an infrared camera from 93 individuals with the purpose of finding the optimal position for temperature sensors. To analyze if this condition affects temperature results, the data was obtained before and after a 100 m walk. After a statistical analysis, the model inferred from it lead us to define five specific areas where the temperature must be measure: heel, medial midfoot, first metatarsal head, fifth metatarsal head and first toe.

(ii) After that, the best type of sensor was also analyzed, since many of them are not adequate for a diabetic foot.

A prototype of a smart sock was developed, with NTC sensors situated in those positions. This device gather a large amount of data with the support of an Arduino board, and then they are transferred and stored in a mobile device for subsequent processing.

After a description of the different types of sensors, the NTC sensor has seemed very good to us to make the prototype. This sensor is characterized by being very small, easy to take measurements, easy to attach to the sock and being the least annoying as it is the smallest. This sensor has many advantages over the others, perhaps the only drawback is its precision, but the precision in this study is not very relevant since we only need to check if the temperature varies in the area of the foot that we are measuring. It could become annoying if the sock used is very thin but being a medium sock there are no discomforts, since the sensor and the threads are integrated with the fibers of the sock. The effects of foot moisture are not taken into account in this first prototype; we are already seeing if significant temperature variations can be seen in the foot.

The development of a sock to measure the temperature of different parts of the foot has been carried out on the basis of a commercial sock, in which electrical wires of a very small section have been inserted so that they will not disturb when the user is moving. The threads are braided with the sock, that is, they are sewn so that they do not move, finally the sensor is also sewn and glued to the sock fabric. The sensors were welded to the electric wire and the weld was covered with shrink material to avoid damaging the foot as much as possible, an objective achieved by this method. All of this gives the prototype a long-lasting consistency and hold so it can be used in data collection sessions. Some tests have already been done, and the data collection has been acquired during one or two hours depending on the disposition of the individuals. Although the sock has worked properly during these sessions, there were some first measurements in which we had to make corrections in the hold of the threads and sensors, but once corrected, the sock measured perfectly. One thing that has not been taken into account was the possibility that foot sweated influenced the measurement; this was discarded since it is not necessary to have a good temperature measurement but to obtain a slight variation in temperature in the areas of the foot studied. For the purposes of this device, a degree or two more or less in the temperature measurement do not influence the results, but what is needed is the appreciation of a temperature change in those points, for this reason the measurements must not be exactly precise either, and hence we have used those tiny sensors that do not harm the patient when they are walking.

The present study has some limitations: (1) The temperature evaluation in this study was recorded only after a 100 m walk, being a short-term assess. It is necessary to re-evaluate the data in a longer walk, to assess its reliability in an activity similar to that of daily life; (2) the solution was tested indoors, so the results (where environmental temperature can affect the body temperature) cannot be extrapolated to outdoors conditions; and (3) the comfort of these smart socks (thinner sensors and wires) to be used by elders in a normal activity in daily life wear must be improved. This issues will be taken into account in future research, where the prototype will be tested in different conditions, looking for better solutions in order to achieve the best wearable to continually assess foot temperature and try to identify early disorders that could avoid infections, plantar ulcers and further amputations in diabetics.

## Figures and Tables

**Figure 1 sensors-21-01821-f001:**
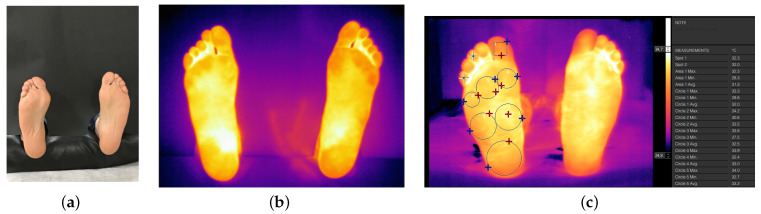
(**a**) Feet covered with a black cardboard prior to the thermal picture, (**b**) Thermographic picture obtained and (**c**) Taking data temperatures of the nine regions of the foot (with Spanish format, where the comma indicates decimal point).

**Figure 2 sensors-21-01821-f002:**
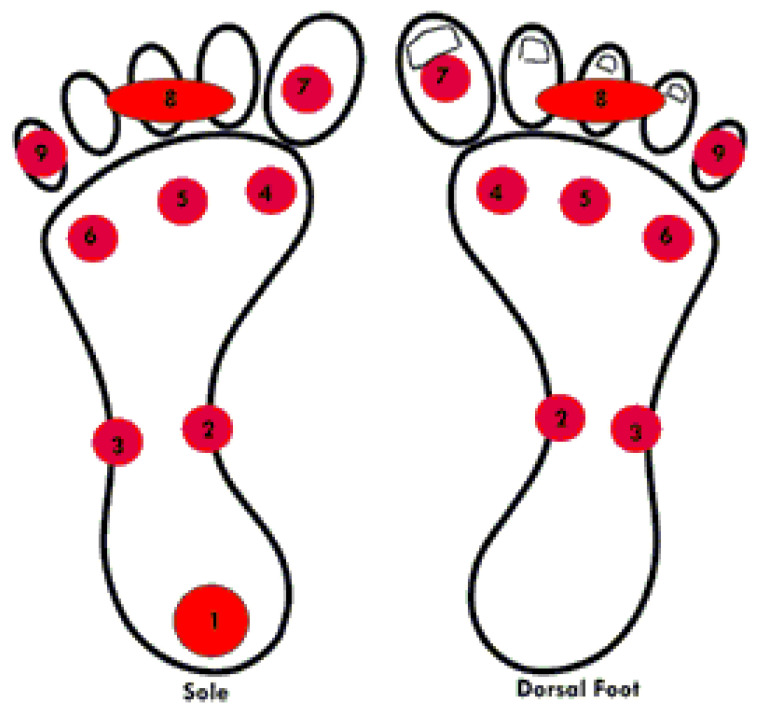
Feet regions where temperature data is collected.

**Figure 3 sensors-21-01821-f003:**
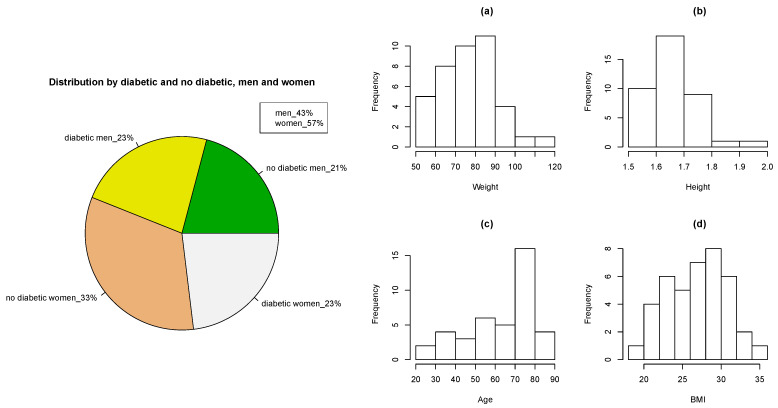
Data distribution (**left**) and Histograms for different variables (**right**): (**a**) Weight, (**b**) Height, (**c**) Age, and (**d**) BMI.

**Figure 4 sensors-21-01821-f004:**
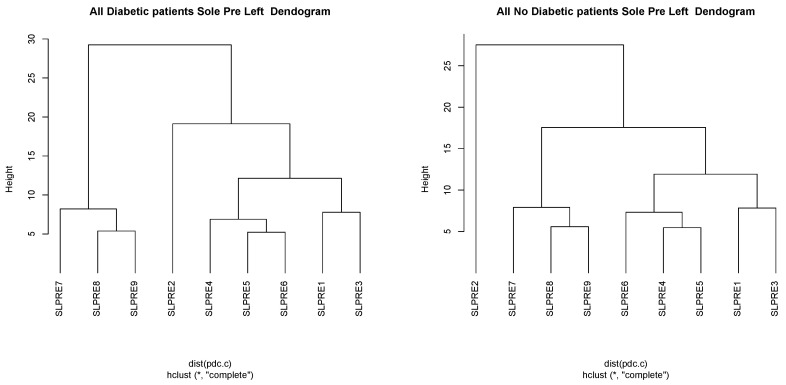
Dendograms of the right sole after the walk: Diabetics (**left**) and nondiabetics (**right**). SLPRE${}_{i}$ represents the temperature for left sole before the walk for index *i*.

**Figure 5 sensors-21-01821-f005:**
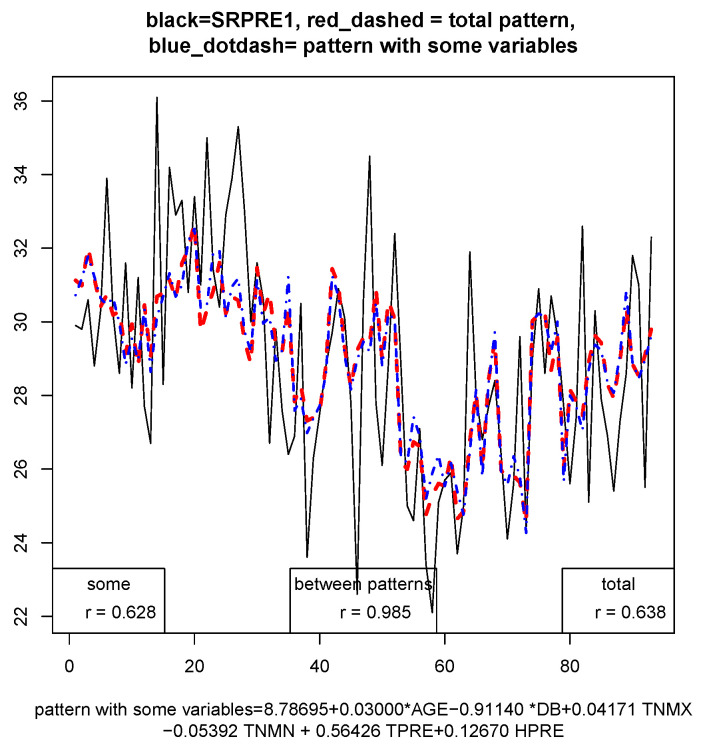
Pattern for the sole of the index 1, right foot before the walk.

**Figure 6 sensors-21-01821-f006:**
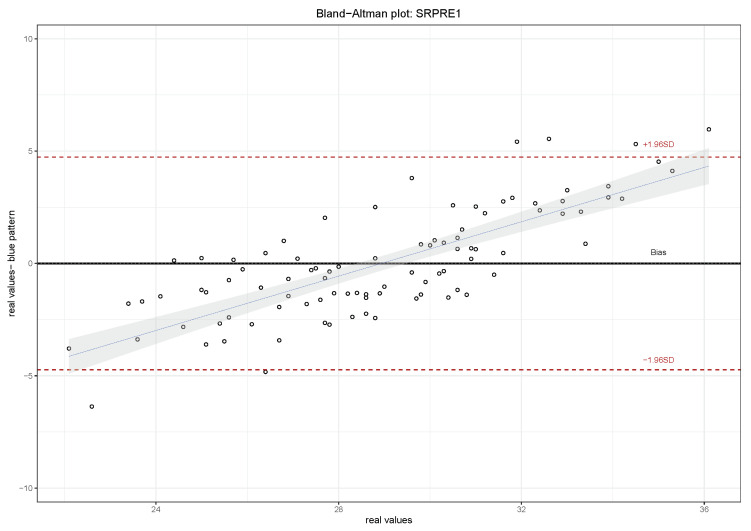
Bland Altman plot, for the sole of the index 1, right foot before the walk, comparing the two measurements: the sample data given by the survey and the model only considering the variables that influence (see Table 1).

**Figure 7 sensors-21-01821-f007:**
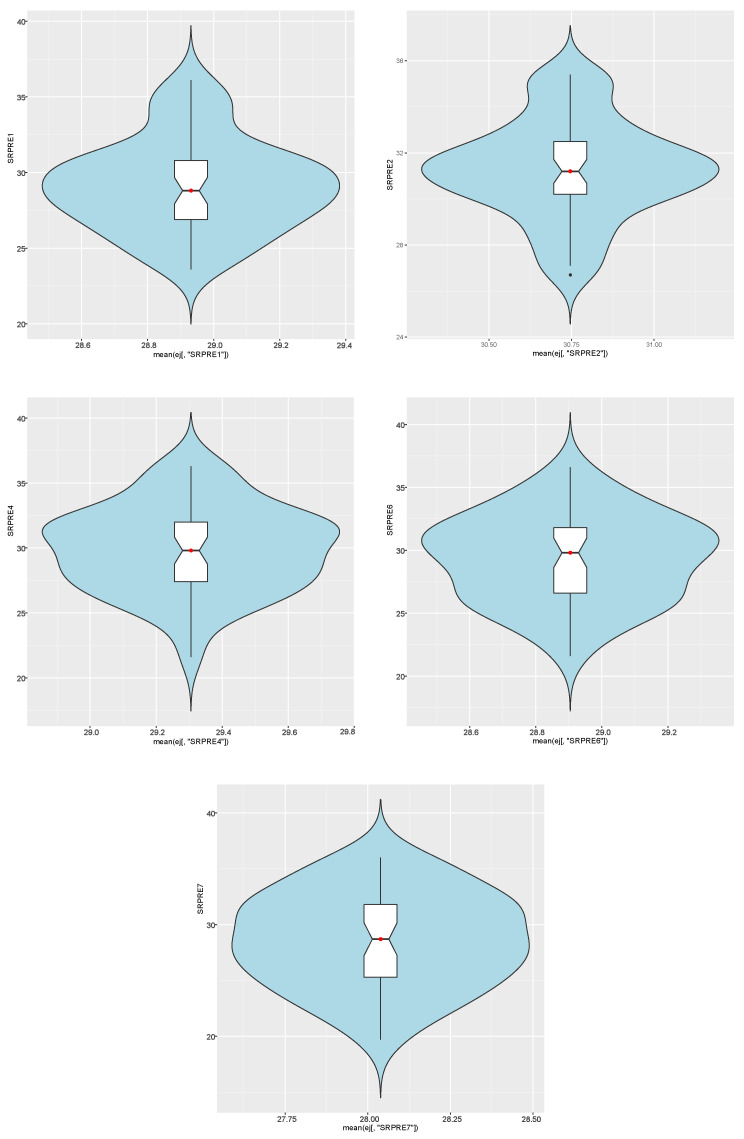
Violin plot for indices $I_{j}$, j∈{1,2,4,6,7}.

**Figure 8 sensors-21-01821-f008:**
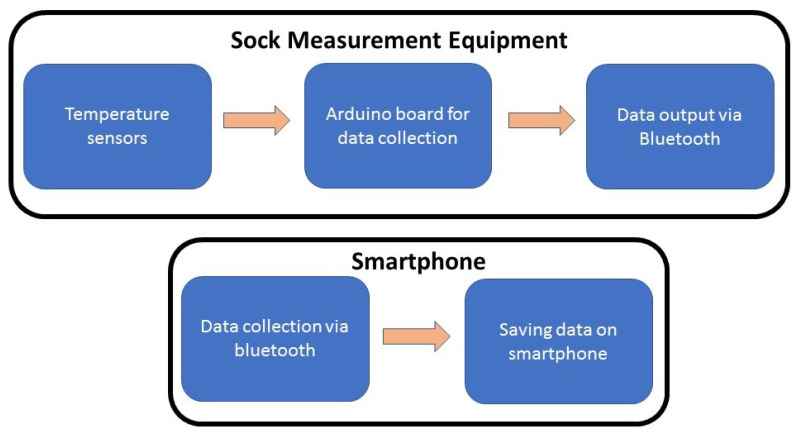
Block diagram of the proposed system that will be able to measure temperature, analyze it and send an alarm to the smartphone when needed.

**Figure 9 sensors-21-01821-f009:**
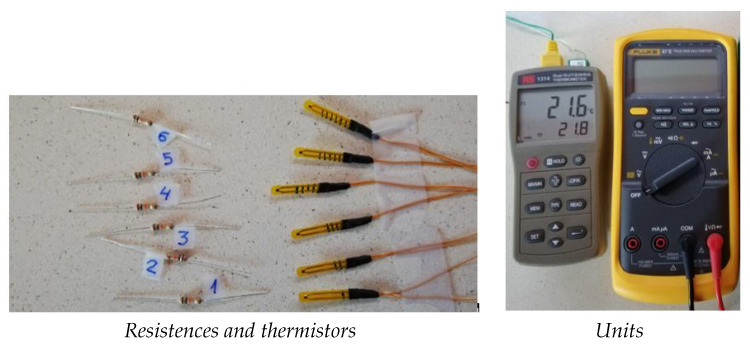
Units calibrated to take the temperature and measure the resistance.

**Figure 10 sensors-21-01821-f010:**
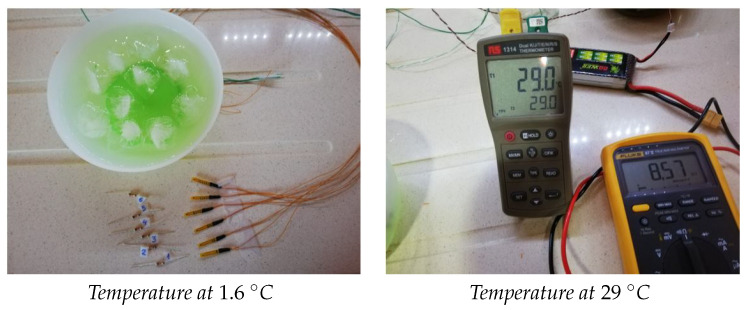
Calculating the parameters of the Steinhart–Hart equation.

**Figure 11 sensors-21-01821-f011:**
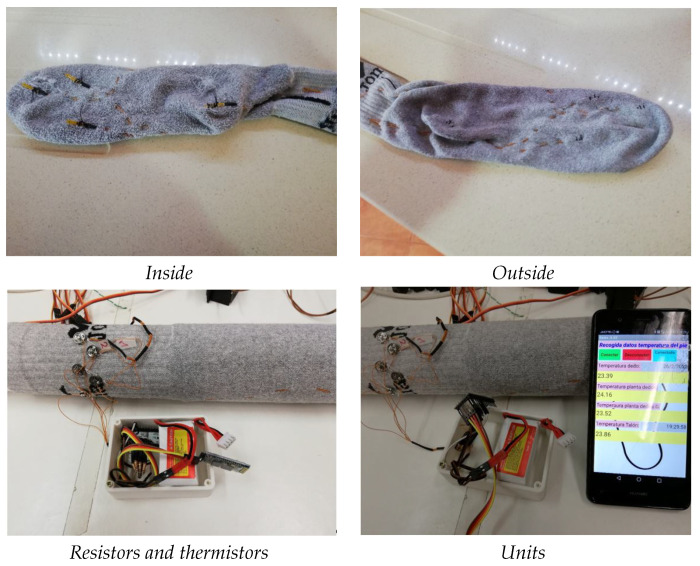
Smart sock working. In the top part we show details of one of the socks with the sensors. In the bottom, readers can see the smart sock working connected to the cell phone application.

**Table 1 sensors-21-01821-t001:** Coefficients that have been obtained with the model. SRPREi represents the temperature for right (R) sole (S) before (PRE) the walk for index i, SLPOSTi represents the temperature for left (L) sole (S) after (POST) the walk for index i, and SRPOSTi represents the temperature for right sole after the walk for index i. Where I can be 1, 2, 4, 6 and 7 indicating the region where the index is considered.

CAT	CTE	AGE	BMI	DB	TNMX	TNMN	TC	TEXT	TPRE	HPRE
SRPRE1	8.78695	0.03		−0.91140	0.04171	−0.05392			0.56426	0.12670
SRPRE2	17.94235	0.01558		−1.47390	0.03612	−0.03644			0.38426	0.07408
SRPRE4	−20.96860	0.02549	0.10193	−1.36591		−0.04557	0.82781		0.67369	0.13880
SRPRE6	−26.14073		0.13748	−1.02450	0.04952	−0.09264	0.79693		0.74866	0.18130
SRPRE7	−38.27672	0.03712	0.11602	−1.28288		−0.05528	1.14328		0.81583	0.15686
SLPRE1	−14.00475	0.03628		−0.84698	0.04798	−0.05280	0.61893		0.53951	0.11614
SLPRE2	15.65169			−0.74912	0.04795	−0.04274			0.39359	0.09685
SLPRE4	−31.30703		0.11339	−1.42929	0.06371	−0.08094	0.93555		0.72724	0.16288
SLPRE6	−32.83695	0.02528	0.09846	−1.33277	0.05504	−0.07547	0.92974		0.76525	0.15994
SLPRE7	−39.72836	0.04965		−1.20896			1.11495		0.84074	0.14443
SRPOST1	−7.01999	0.04641		−0.93266			0.48558		0.64667	
SRPOST2	−3.44379		0.10557	−1.33277	0.06073	−0.04858	0.56932	0.07686	0.27156	
SRPOST4	−15.45718	0.02753		−1.63835	0.04549	−0.05258	0.67605		0.73318	
SRPOST6	−20.78108		0.11620	−1.07018	0.05828	−0.08463	0.65984		0.79016	0.08394
SRPOST7	−24.79945	0.03964		−1.35335			0.88517		0.78046	
SLPOST1	−12.99850	0.03859		−1.03925	0.05417	−0.04706	0.58444		0.63557	
SLPOST2	15.67648	0.02831		−1.21562	0.03380				0.39475	
SLPOST4	−15.34189	0.02699		−1.32829	0.04989	−0.05053	0.62868		0.74446	
SLPOST6	−20.94168	0.02664		−1.15666	0.05514	−0.04966	0.73794		0.76450	
SLPOST7	−29.61293	0.04579		−1.07319			0.96990		0.81785	

**Table 2 sensors-21-01821-t002:** Difference between the indices on the two sole (left and right) for nondiabetic people before the walk in the regions 1, 2, 4, 6 and 7 (see Figure 1). Here we can see the most important statistic of then like mean, the standard deviation (sd), the standard error of the mean (sem), interquartile range (IQR), the coefficient of variation (cv), the degree of distortion from the symmetrical bell curve or the normal distribution or the measure of symmetry (Skewness), the measure of whether the data are heavy-tailed or light-tailed relative to a normal distribution (Kurtosis), and the quartiles.

DSPRE	Mean	sd	se(mean)	IQR	cv	Skewness	Kurtosis	0%	25%	50%	75%	100%
DSPRE1	0.0938	0.8234	0.1176	0.7	8.7716	−1.7464	7.6044	−3.6	−0.2	0.1	0.5	1.7
DSPRE2	0.4326	0.5550	0.0792	0.6	1.2828	0.2910	0.9031	−0.9	0.1	0.4	0.7	2.0
DSPRE4	0.2081	1.2282	0.1754	0.9	5.9005	−1.1939	3.8264	−3.9	−0.1	0.3	0.8	3.1
DSPRE6	0.0551	0.9757	0.1393	1.1	17.7082	−1.2871	5.0223	−4.0	−0.5	0.1	0.6	1.8
DSPRE7	0.1163	1.1422	0.1631	1.1	9.8193	0.25053	0.3911	−2.7	−0.5	0.0	0.6	3.0

**Table 3 sensors-21-01821-t003:** Data of resistances (Res 1 to 6) and temperatures (Temp 1 to 6) taken by the six NTC sensors.

Resistence/Thermistor	Res 1	Temp 1	Res 2	Temp 2	Res 3	Temp 3	Res 4	Temp 4
R1/NTC1	9.80 kΩ	$21.6{\;}^{\circ }$C	24.88 kΩ	$1.6{\;}^{\circ }$C	8.31 kΩ	$29.7{\;}^{\circ }$C	5.02 kΩ	$43.5{\;}^{\circ }$C
R2/NTC2	9.83 kΩ	$21.6{\;}^{\circ }$C	24.33 kΩ	$1.6{\;}^{\circ }$C	8.38 kΩ	$29.7{\;}^{\circ }$C	5.20 kΩ	$43.5{\;}^{\circ }$C
R3/NTC3	9.82 kΩ	$21.6{\;}^{\circ }$C	24.19 kΩ	$1.6{\;}^{\circ }$C	8.30 kΩ	$29.6{\;}^{\circ }$C	5.15 kΩ	$43.3{\;}^{\circ }$C
R4/NTC4	9.85 kΩ	$21.6{\;}^{\circ }$C	24.50 kΩ	$1.6{\;}^{\circ }$C	8.34 kΩ	$29.4{\;}^{\circ }$C	5.19 kΩ	$43.2{\;}^{\circ }$C
R5/NTC5	9.79 kΩ	$21.6{\;}^{\circ }$C	25.12 kΩ	$1.6{\;}^{\circ }$C	8.34 kΩ	$29.4{\;}^{\circ }$C	5.26 kΩ	$42.8{\;}^{\circ }$C
R6/NTC6	9.84 kΩ	$21.6{\;}^{\circ }$C	25.21 kΩ	$1.6{\;}^{\circ }$C	8.47 kΩ	$29.3{\;}^{\circ }$C	5.47 kΩ	$42.0{\;}^{\circ }$C

**Table 4 sensors-21-01821-t004:** Parameters of the Steinhart–Hart equation for the six NTC sensors. These parameters *A*, *B* and *C* are constants that are obtained for the equation that defines the resistance variation of the sensors. *B* were multiplied times $10^{3}$, *B* times $10^{4}$ and *C* times $10^{7}$.

Thermistor	$A\cdot 10^{3}$	$B\cdot 10^{4}$	$C\cdot 10^{7}$
NTC1	1.409294790	1.684156947	5.069508025
NTC2	1.149226326	2.051360360	4.064329590
NTC3	1.076255140	2.180997134	3.519982960
NTC4	0.762209845	2.695628622	1.483573950
NTC5	1.300653498	1.843001087	4.536745780
NTC6	0.579418628	2.997765865	0.2115108092

## Data Availability

Not applicable.

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
