# Peer review of "The Use of Infrared Thermography to Develop and Assess a Wearable Sock and Monitor Foot Temperature in Diabetic Subjects"

_sensors, 2021, doi:10.3390/s21051821_

Round 1

Reviewer 1 Report

This is an interesting study focusing on the important health care topic. However, the results and performance indices must be reported in a rigorous way to support the conclusion.

The entire tables such as table 1 must be self explanatory.

When regression is used, the goodness of fit must be presented in R Square with CI 95%.

Also, the Bland Altman plot is missing.

In the ststsital method, no proper statistical method was introduced.

Also, the residual analysis of the regression is missing.

How was the sample size estimated ?

Author Response

Answer to Reviewer 1:

This is an interesting study focusing on the important health care topic. However, the results and performance indices must be reported in a rigorous way to support the conclusion.

The entire tables such as table 1 must be self explanatory.

We have changed the captions of the tables to do them self explanatory.  Thank you.

When regression is used, the goodness of fit must be presented in R Square with CI 95%.

We are guessing that he is talking about figure 4.  In this case, reviewer 1 is totally right.  Those models (with all the variables, and only with selected variables) are not predictive.  We think that this is because we were taking temperatures with the infrared camera punctually, and it might not consider important factors.

This is one of the reasons why we wanted to take temperatures in a more continuous way to obtain more information.

Also, the Bland Altman plot is missing.

Also, the residual analysis of the regression is missing.

We have now included a Bland Altman plot comparing the model with selected variables, with the sample obtained in the survey in the index 1, before the walk.  Results were similar for other indices.

In the statistical method, no proper statistical method was introduced.

Now, we introduced the different statistical methods that we utilized in the paper.

We have used the Stepwise command (with the option Stepwise selection, with both forward and backward) given by R without predictors.  We have now included a sentence about it in the text.  Thank you.

How was the sample size estimated ?

This is an important, but also a difficult question.  In our case, we were only able to obtain data from one institution located in Plasencia, a not very big Spanish town.  If we had only wanted to analyze the comfort of our patients, then we would have taken a much smaller sample size.

Since we would like to continue with the work and analyze the comfort of people from other locations and after longer walks or doing moderate exercise, and also because we are considering a good number of variables in our study, then we decided to use a size large enough (almost 100 patients).  Actually, we are considering adding more patients in the next future.

Reviewer 2 Report

  1. Positive - This seems quite a novel study to detect the variation in temperature in foot for diabetic patient. Finding ROI accurately can be really impactful for these cases.
  2. Positive - Smart sock has been evaluated on good size of data to test the feasibility.
  3. Positive - Results have been presented well.
  4. Negative - Although the problem is important, the solution seems pretty straight forward. It is not very clear why this solution is unique.
  5. Negative - Authors have not tested their solution in outdoors where environmental temperature can affect the body temperature. This can be a real deal breaker. 
  6. Negative - How comfortable these smart socks to wear and to maintain by elders have not been discussed.

Author Response

 Answer to Reviewer 2:

Positive - This seems quite a novel study to detect the variation in temperature in foot for diabetic patient. Finding ROI accurately can be really impactful for these cases.

Positive - Smart sock has been evaluated on good size of data to test the feasibility.

Positive - Results have been presented well.

Negative - Although the problem is important, the solution seems pretty straight forward. It is not very clear why this solution is unique.

The solution is not unique. We propose a solution to measure the temperature of the foot according to temperature zones chosen according to scientific criteria, with data that support this choice. In the previous literature, the temperature zones studied were chosen without this criterion, or it was not specified how they had been chosen.

We believe that to know the most suitable areas to place the sensors can make our solution optimal for effective temperature monitoring and help avoid the risks associated with these types of feet.

Negative - Authors have not tested their solution in outdoors where environmental temperature can affect the body temperature. This can be a real deal breaker.

The solution we provide has not been carried out outside, the measures have been in a controlled place, specifically inside a building at the University of Extremadura and although it was maintained for several hours, the process of taking measures was correct. As we present a prototype, its operation was tested with the conditions describer in the study. One of the problems that we found was the disconnection of the sensors due to the movement of the foot, something that can be solved with better welding and even with the assembly of conductive technical textiles.

Thanks for your comment. In line 441 we have included that this is a prototype.

Negative - How comfortable these smart socks to wear and to maintain by elders have not been discussed.

As it is a prototype, we first tested it checking that its adaptation to the foot was good and did not cause discomfort. For its connection with the mobile, it is quick and easy to perform; the data obtained is accumulated in a file easy to process. We have checked several sensors and the most comfortable is the selected one.

------

Thanks for your valuable comments. With the modifications included in the new version, all aspects of the prototype are more clear.

Reviewer 3 Report

This article seeks to describe a framework for monitoring temperature in diabetic foot. The overall article is of interest; however, I have previously commented and rejected this article due to fundamental concerns surrounding the methodology, duplication of previous works and inability to reproduce the results. The authors have not provided any formal response to the final round of peer review comments and have also only made minor changes to the manuscript (when performing a compiarson between both submissions).

As with my previous comments, the paper does not meet the technical quality required to be considered for publication in MDPI Sensors. Specific comments hereafter:

  1. The authors should consider revising the title to reflect the methods and techniques used in the manuscript. Provide specifics of the methodology used.
  2. The structure of the paper is difficult to follow. The authors should consider revising the manuscript to ensure that the framework is presented in the methods section, but results are presented in the results. An example would be:
    1. Introduction
    2. Background
    3. Methods (framework, analyses, proposed approach, app development)
    4. Results (including location selection, validation and testing)
    5. Conclusions and future work
  3. Abstract: Overall the abstract is well structured, however more specificity will help the read. For example, what benefit would regular screening and education have on reducing cases of diabetic foot? His question is raised as there is no introduction to what diabetic foot is in the first sentence.
  4. Introduction: The introduction would benefit from further structure. At the moment the authors provide little overview of diabetic foot, and instead jump straight into discussing temperature and stocks. Perhaps something along the lines of:
    1. Diabetic foot – what is it, how much of a problem is it, how many deaths, comorbidity;
    2. The role temperature (and other aspects such as pressure points and activity) has on diabetic foot.
    3. The utility of wearables, such as socks, in support those with diabetic foot and detecting temperature.
    4. Overall aims of the paper.
  5. Methods:
    1. You need to provide justification as to why each data point was collected, and how it will be used in the analysis.
    2. Did the authors use any validated measures to assess habits and comforts?
    3. How were participants recruited?
    4. Figure 2. Did the authors identify any differences between the groups? It would be useful to provide a table of these recruits, as well standard deviation or confidence intervals, as an appendix item.
  6. Mobile telephone: The authors introduce the mobile telephone in the results section yet provide little information about the framework. It would be useful to provide a process diagram of the whole system and how this mobile phone is used. However, this then appears in Section 4.
  7. How long did it take the research team to ‘set up’ for each participant?
  8. Please confirm why Excel has been selected, is the file format .csv, if so, please specify this.

These comments are broad and largely replicate my previous review.

Author Response

Answer to Reviewer 3:

Specific comments hereafter:

  1. The authors should consider revising the title to reflect the methods and techniques used in the manuscript. Provide specifics of the methodology used.

Following the Reviewer’s suggestion, we have changed the title to "The use of infrared thermography to develop and assess a wearable sock and monitoring foot temperature in diabetic subjects", which reflex the technique used.  Thank you

  1. The structure of the paper is difficult to follow. The authors should consider revising the manuscript to ensure that the framework is presented in the methods section, but results are presented in the results. An example would be:
    1. Introduction
    2. Background
    3. Methods (framework, analyses, proposed approach, app development)
    4. Results (including location selection, validation and testing)
    5. Conclusions and future work

We would like to thank reviewer’s comment.  However, we consider that this structure (proposed by the Reviewer) is not the most suitable for this type of work. 

First of all, we wanted to show how the evolution of our research was developed, and, for example, it does not help to include the location selection at the end of the paper.  Actually, for the elaboration of the sock, we explained in the text that we needed to select the location of the sensors at the beginning of our work.  Readers might be confused if, for example, the app development (which is one of the latest steps to take) appears before the location selection.

Also, since this work is interdisciplinary, it does not help to many possible readers who are looking for solutions / devices for similar problems, because the proposed structure mix all the proposed approach.

  1. Abstract: Overall the abstract is well structured, however more specificity will help the read. For example, what benefit would regular screening and education have on reducing cases of diabetic foot? His question is raised as there is no introduction to what diabetic foot is in the first sentence.

We have now changed the abstract following Reviewer’s suggestion.  And a sentence explaining what a diabetic foot syndrome has been added.

  1. Introduction: The introduction would benefit from further structure. At the moment the authors provide little overview of diabetic foot, and instead jump straight into discussing temperature and stocks. Perhaps something along the lines of:
    1. Diabetic foot – what is it, how much of a problem is it, how many deaths, comorbidity;
    2. The role temperature (and other aspects such as pressure points and activity) has on diabetic foot.
    3. The utility of wearables, such as socks, in support those with diabetic foot and detecting temperature.
    4. Overall aims of the paper.

We have changed the introduction following Referee’s suggestion, and we divided it in the proposed subsections.

  1. Methods:
    1. You need to provide justification as to why each data point was collected, and how it will be used in the analysis.
    2. Did the authors use any validated measures to assess habits and comforts?
    3. How were participants recruited?
    4. Figure 2. Did the authors identify any differences between the groups? It would be useful to provide a table of these recruits, as well standard deviation or confidence intervals, as an appendix item.

5.1. An explanation of why the 17 ROIs were selected has been added.

5.2. No, as the primary aim of the study was to assess regions more specifically and to develop the sock, the habits and comforts were asked following an own questionnaire.

5.3. All patients belong to the CPUEX clinic of the University, and a sentence has been added explaining it.

5.4. There were no significative differences between the two groups.

  1. Mobile telephone: The authors introduce the mobile telephone in the results section yet provide little information about the framework. It would be useful to provide a process diagram of the whole system and how this mobile phone is used. However, this then appears in Section 4.

  1. How long did it take the research team to ‘set up’ for each participant?

Each patient takes 20 minutes to assess it. This sentence has been added in the text.

  1. Please confirm why Excel has been selected, is the file format .csv, if so, please specify this.

Yes, the file format is .csv, and this text has been added in the text.

Round 2

Reviewer 1 Report

The authors considered most of the issues addressed by the viewer. However,  some of the important topics were not completely taken into account. I also provided some minor improvement details prior to publication Thus, I invite the authors to consider the following issues. They must be considered so that the paper will be suitable for publication.

Major issues:

As in the data collection, 93 subjects participated in the study, and no sample size calculation method was provided, this must be mentioned as a limitation in the discussion. This must be discussed along with the proper guidelines such as STARD.

As far as I understood, no random sampling was used and the sampling method is non-random. This is another important issue that must be mentioned in the discussion as a limitation.

Cases such as lines 302, 303, where the correlation was provided must be accompanied by their CI 95%. It showed the reliability of the estimates.

Fig.4, right, It is impossible to read the captions of the figure ... x and y labels, any text inside the figure ... it must be improved. Also, the legend of the left panel has overlap with the figure that must be corrected.

Based on the BA plot (Fig.4 right), the method has a clear bias that is dependent on the higher or lower values of the quantity of interest. It R-square must be provided and also some points are out of the upper limit line. All of these issues must be mentioned as the limitation of the method.

Minor issues,

In the abstract, at least one quantitative performance measure must be provided.

Table 4, 1e-3,1e-4 and 1e-7 must be factored in indices A,B, C to avoid repetions e.g., A (* 1e-3).

Author Response

Answer to Reviewer 1:

As in the data collection, 93 subjects participated in the study, and no sample size calculation method was provided, this must be mentioned as a limitation in the discussion. This must be discussed along with the proper guidelines such as STARD.

We have included a new paragraph at the beginning of subsection 3.1 following Reviewer’s suggestion.

As far as I understood, no random sampling was used and the sampling method is non-random. This is another important issue that must be mentioned in the discussion as a limitation.

We have included a paragraph and a citation.  Thank you.

Cases such as lines 302, 303, where the correlation was provided must be accompanied by their CI 95%. It showed the reliability of the estimates.

Thanks, we have included the information of the CI.

Fig.4, right, It is impossible to read the captions of the figure ... x and y labels, any text inside the figure ... it must be improved. Also, the legend of the left panel has overlap with the figure that must be corrected.

We have separated both parts of previous Figure 4, we think that it is easier to check captions now.

Based on the BA plot (Fig.4 right), the method has a clear bias that is dependent on the higher or lower values of the quantity of interest. It R-square must be provided and also some points are out of the upper limit line. All of these issues must be mentioned as the limitation of the method.

Thanks for the comment. We have included the value of R^2 = 0.3948 and some remarks about the model.

In the abstract, at least one quantitative performance measure must be provided.

It is done.

Table 4, 1e-3,1e-4 and 1e-7 must be factored in indices A,B, C to avoid repetions e.g., A (* 1e-3).

Done.  Thank you

Reviewer 3 Report

Editor comments provided. 

Author Response

We are sorry, but we did not see any suggestion for changes from Reviewer.  We are open to discuss proposals to improve the paper